# *Pythium* Damping-Off and Root Rot of *Capsicum annuum* L.: Impacts, Diagnosis, and Management

**DOI:** 10.3390/microorganisms9040823

**Published:** 2021-04-13

**Authors:** Himanshu Arora, Abhishek Sharma, Satyawati Sharma, Farah Farhanah Haron, Abdul Gafur, R. Z. Sayyed, Rahul Datta

**Affiliations:** 1Centre for Rural Development and Technology, Indian Institute of Technology, New Delhi 110016, India; himanshuarora592@gmail.com (H.A.); satyawatis@hotmail.com (S.S.); 2Amity Food and Agriculture Foundation, Amity University, Noida 201313, Uttar Pradesh, India; 3Pest and Disease Management Program, Horticulture Research Center, Malaysian Agriculture Research and Development Institute (MARDI), Persiaran MARDI-UPM, Serdang 43400, Selangor, Malaysia; farahfarhanah@mardi.gov.my; 4Sinarmas Forestry Corporate Research and Development, Perawang 28772, Indonesia; gafur@uwalumni.com; 5Department of Microbiology, PSGVP Mandal’s Arts, Science, Commerce College, Shahada 425409, Maharashtra, India; 6Department of Geology and Pedology, Mendel University in Brno, 613 00 Brno-sever-Černá Pole, Czech Republic

**Keywords:** control strategies, diagnosis, epidemiology, post-emergence damping-off, pre-emergence damping-off, *Pythium*, resistant cultivars, root rot

## Abstract

*Capsicum annuum* L. is a significant horticulture crop known for its pungent varieties and used as a spice. The pungent character in the plant, known as capsaicinoid, has been discovered to have various health benefits. However, its production has been affected due to various exogenous stresses, including diseases caused by a soil-borne pathogen, *Pythium* spp. predominantly affecting the *Capsicum* plant in younger stages and causing damping-off, this pathogen can incite root rot in later plant growth stages. Due to the involvement of multiple *Pythium* spp. and their capability to disperse through various routes, their detection and diagnosis have become crucial. However, the quest for a point-of-care technology is still far from over. The use of an integrated approach with cultural and biological techniques for the management of *Pythium* spp. can be the best and most sustainable alternative to the traditionally used and hazardous chemical approach. The lack of race-specific resistance genes against *Pythium* spp. can be compensated with the candidate quantitative trait loci (QTL) genes in *C. annuum* L. This review will focus on the epidemiological factors playing a major role in disease spread, the currently available diagnostics in species identification, and the management strategies with a special emphasis on *Pythium* spp. causing damping-off and root rot in different cultivars of *C. annuum* L.

## 1. Introduction

Cultivars of *Capsicum annuum L.* (Solanaceae) are an essential part of many cuisines worldwide as a spice. *C. annuum* L. is a semi-perennial herbaceous plant that is usually grown as an annual crop worldwide, including in India [1]. The genus *Capsicum* is split into three gene complexes based on the crossing ability between species [2], of which the species of *C. annuum* complex (*C. annuum*, *C. frutescens,* and *C. chinense*) is commercially most important [3]. In 2018, worldwide production of dry chilies and peppers was estimated to be up to almost 4.2 million tons over an area of 1.8 million hectares. India is the biggest producer of dry chilies and pepper grown worldwide. Capsaicinoids are distinctive of the *Capsicum* genus and are produced entirely inside the fruit placenta [4], which causes the characteristics of fruit pungency in this genus [5]. Capsaicinoids are a mixture of at least 23 alkaloids (vanillylamines), a major representative of which are dihydrocapsaicin and capsaicin, which collectively account for almost 90% of the total capsaicinoid content in the fruit [6]. A mildly or non-pungent analog of the capsaicinoids, namely capsinoid, is also present in pungent varieties in trace amounts and shares a similar structure with capsaicinoids [7]. Fresh *Capsicum* fruits are also rich in phenolic content, including flavonoids, phenolic acids, and tannins [8]. Mature fruits are abundant in carotenoids such as capsanthin, capsorubin, β-carotene, etc., of which capsanthin is the major contributor [9]. Minerals such as potassium, phosphorous, magnesium, calcium, sodium, iron, manganese, boron, selenium, copper, and zinc are commonly found in *Capsicum*; however, their content is dependent on variables such as the fruits’ variety, growth stage, environmental factors, and the cultivation practices [10]. 

Various health benefits have been found to be related to different *Capsicum* cultivars. Fresh green peppers and red peppers have ample amounts of vitamin A, C, and antioxidant compounds [6,11,12,13]. Srinivasan [14], in his review on the biological activities of capsaicin, documented the various potential anti-carcinogenic, anti-inflammatory, pain relief, weight loss, gastrointestinal, and cardioprotective effects of capsaicin in detail. 

Small-fruited cultivars were observed to form a more divergent phylogenetic group than the large-fruited varieties, making the small-fruited cultivars genetically more distant [15]. Although various breeding programs have led to the creation of cultivars with favorable traits, this has still not found full application in making disease-resistant varieties because of many interspecific crossing barriers [6,16], including many factors both pre and post-fertilization [2]. Production of *C. annuum* is restricted by many fungal, bacterial, viral, and nematode diseases associated with it worldwide [17]. The damping-off caused by various *Pythium* spp. is a significant disease of *C. annuum* cultivars that predominantly occurs in nursery beds affecting seeds and young seedlings [18,19]. When the plant is in the seed or seedling stage of its life cycle, infection by *Pythium* species causes pre-emergence and post-emergence damping-off, decaying the seeds and seedlings before emergence and after the emergence of the plant from the soil surface, respectively. However, infected mature plants have also been found to show root rot symptoms. *Pythium* spp.mainly affect the younger or juvenile tissues, which have not yet developed any secondary thickenings; thus, the infection is limited to seeds, seedlings, and the younger roots. The post-emergence damping-off of seedlings is often associated with symptoms such as reduced growth, water soaking, wilting, black or brown discoloration, and root rot [20,21,22]. In more mature plants, water-soaked roots and lesions of stem at the soil line, stunted growth, and brown discoloration of roots are prevalent [23]. 

In India, chili was first reported as the host of *Pythium* spp. over 100 years ago [24]. Since then, there have been repeated records of damping-off and root rot incidences caused by *Pythium* spp. in chili [25,26]. Pre-emergence and post-emergence incidences ranging from 7% to 90% of crops have been reported in several states of India [27,28,29,30,31]. In Pakistan, many occurrences of damping-off and root rot disease incidences ranging from 13% to 46% have been reported [20,22,32], and an estimated loss of Rs. 70,000–100,000/acre, in the case of hot pepper, has been observed [33]. Aside from damaging crops directly, *Pythium* spp. have also been observed to break resistance to nematodes in chili plants [34]. Financial losses due to *Pythium* infections are not limited to direct damage to crops; instead, they also include the re-sowing costs [35].

Although *Pythium* spp. has been found to infect both sweet and pungent varieties, the current study is centered on the pungent cultivars. Figure 1 presents a brief summary of the components reviewed in this study. This study focuses on the techniques that have been followed to detect and diagnose *Pythium* spp., the management strategies studied to control *Pythium* spp.-related diseases in *Capsicum* since 2010, and resistant breeding against *Pythium* spp. 

## 2. *Pythium* Species: Causal Agent of Damping-Off and Root Rot

The genus *Pythium* is a part of the Peronosporales order of the class Oomycetes or the phylum Oomycota, a member of the kingdom Chromista (Stramenopiles) [36]. The *Pythium* spp. are no longer considered to be true fungi; instead, they were found to be more closely related to algae [37]. The class Oomycetes differ from true fungi in that they have coenocytic hyphae, a diploid vegetative stage, cell wall content of β-glucan and cellulose, biflagellate zoospores with tinsel, and whiplash flagella [37,38]. The *Pythium* genus contains species that are pathogenic to plants [39,40,41], animals [42], algae [43], and other fungi [44]. More than 330 species of *Pythium* have been identified to date (www.mycobank.org), of which many are pathogenic to plants. These pseudo-fungi cause many plant diseases, including damping-off, root rot, collar rot, soft rot, and stem rot in different production systems such as nurseries, greenhouses, and agricultural fields [45,46,47]. In *C. annuum* plants, many *Pythium* spp. have been observed to cause diseases. Some of these species have been identified using the morphological keys available and gene sequences (Table 1). The next few sections discuss the ecology of the pathogen involved and the critical factors involved in disease epidemics on *C. annuum* cultivars caused by various *Pythium* spp. 

### 2.1. Ecology

*Pythium* spp. are soil-borne fungi. Indeed, members of this genus are no longer considered to be fungi but still share a striking similarity with true fungi in having vegetative (mycelium), asexual (sporangia and zoospores), and sexual (oospores) stages in their life cycle [54]. Figure 2 provides a standard overview of the disease cycle of *Pythium* species in pepper plants. *Pythium* spp.inhabit the soil either as a propagule (Zoospore, oospore, or sporangia) or in the mycelial form. The oospores are the primary survival structure and infecting unit of the many species of the *Pythium* genus in the soil, which can stay viable in the soil for long durations [37,55]. However, sporangia of some *Pythium* species have also been reported to be survival propagules in soil [56,57]. The survivability of propagules of *Pythium* in soil follow two critical mechanisms: (1) the constitutive dormancy of oospores—i.e., their germination requires an internal stimulation other than the external stimulus, because of which all the propagules present in the soil do not germinate simultaneously; and (2) the formation of secondary resting propagules in the case of a shortfall of ample nutrients [55,58]. The propagules can germinate and penetrate the host plant very rapidly in the presence of exudates from roots and other supporting exogenous stimuli, which makes their control very difficult [39]. Sporangia can infect the plant either directly through hyphal tube germination or by producing zoospores in the presence of high moisture conditions [55]. Besides the propagules, *Pythium* species also survive saprophytically as mycelium in the soil and usually colonize the fresh organic substrate. However, this stage of their life cycle is very prone to competition by other soil colonizers [59]. In the seeds, seedlings, and younger roots of mature plants, after penetration, *Pythium* usually survives necrotrophically, thus eventually leading to mortality. 

### 2.2. Epidemiology

*Pythium* spp.are naturally attracted to the exudates released from the germinating seeds, which are stimulatory to the growth of their propagules. While germinating, the seeds imbibe water and release exudates, making them the primary target of infection by a *Pythium* propagule [55]. Exogenous environmental factors have a critical role in the predisposal of plants to infection and disease spread in the host after infection by *Pythium*. However, the favorable conditions for disease spread by *Pythium* tend to vary with the species [60]. Damages ranging from small outbreaks to an epidemic-level scale depend mostly on variables such as moisture, temperature, and organic matter content; however, factors such as pH and soil type also play a crucial role. 

Temperature is undeniably the critical factor in the spread of disease by *Pythium*. An array of temperature ranges represent thriving conditions for various *Pythium* spp. such as *P. graminicola* have been observed to cause more damage in *Capsicum* at high cardinal temperatures [21].

Besides supporting the growth of *Pythium*, high-temperature conditions, in some cases, place additional abiotic stress on the plants, which eventually enhances the severity of root rot [61]. A high moisture content in the soil supports the motility of zoospores and increases the size of the spermosphere—factors that play a crucial role in infections by *Pythium* propagules [55]. Hyder et al. [62] observed high disease incidences and an increased severity of damping-off in chili pepper crops in areas with canal irrigation systems, and Majeed et al. [30] also reported that spaces with high natural moisture were highly infested. A greenhouse environment with damp conditions and high temperatures provides thriving conditions for *Pythium* [63]. Muthukumar et al. [64] observed the highest incidences of damping-off in chili in clay soils, which was attributed to their high moisture-holding capacity. Organic matter content in the soil sometimes supports the saprophytic growth of *Pythium* in the absence of antagonistic microbial communities. Hansen and Keinath [65] observed organic amendments of *Brassica* species that, rather than being suppressive, supported the increase of *Pythium* populations. 

## 3. Detection and Diagnosis of *Pythium* Species

*Pythium* spp. pathogenic to *C. annuum* plants have been identified using morphological features and sequencing marker regions of the DNA of the isolated pathogen. Levesque and Cock [66] found the sporangial morphology to be the most reliable morphological feature in species identification, which was very much consistent with major clades formed in molecular characterization. However, the correct identification of the *Pythium* spp. is often hindered by the inability of heterothallic species to form reproductive structures in cultures, overlapping some characters, and the inability to distinguish intraspecific variation [21,66,67]. Nevertheless, the identification based on morphological features is neither very time-efficient nor very reliable. 

Pathogen detection and diagnosis have a vital role in correlating epidemiological factors with the species involved in the disease spread. Damping-off and root rot diseases sometimes involve a complex of pathogens, including species other than *Pythium* spp. [68]. The early diagnosis of *Pythium* can help in managing the further spread of the disease. The accurate detection of the relevant species can prevent the wasteful usage of pesticides and other resources because different species differ in their susceptibility to control strategies [38]. Thus, the potential diagnostic techniques have the prerequisites of being rapid, accurate, easy to use, and cost-effective. The detection and diagnostic methods currently used for plant pathogens can be broadly divided into two categories; i.e., immunodiagnostic methods and nucleotide-based methods [69]. In the context of *Pythium* spp., recent studies have been primarily focused on advanced PCR-based molecular techniques such as real-time PCR, multiplex PCR, and loop-mediated isothermal amplification (LAMP). The Internal Transcribed Spacer (ITS) region, including ITS 1, 5.8 s, and ITS 2 sequences of nuclear rDNA, is the most frequently used sequence for identifying *Pythium* spp. [20,21,49,70]. The primers developed by White et al. [71], based on some conserved sequences in eukaryotic DNA, as well as the primers based on the ITS sequence of the specific species, are used most frequently in the PCR amplification of the ITS region of the specific species. 

Immunological methods rely on antigen–antibody specificity. Although immunological methods for *Pythium* have been minimal because of the difficulty in producing species-specific antibodies [37], they have a better potential for use in point of care test systems operated by non-specialists [72]. Ray et al. [73,74] used the polypeptides released during fungal infections as immunogens to develop polyclonal antibodies to detect early *P. aphanidermatum* infections in turmeric and ginger rhizomes, respectively. In an ELISA analysis, the developed antibodies showed a good sensitivity at very low concentrations. Although the antigen load in the early stages of infection in rhizome was remarkably low compared to more severely infected rhizomes, the infection was still detectable. 

Similar to the rapid detection of the pathogen, inoculum quantification is also important. In real-time PCR amplification, the PCR product is monitored as it is amplified after each cycle, providing an estimate of the pathogen present in the environmental sample [75]. Based on the inoculum present in the soil and plant, the amount of pesticide application, crop area selection, non-host species, and seasonal variation in the pathogen spread can be determined. Furthermore, the resistant varieties and genetically modified resistant crops can be studied by observing the pathogen count in asymptomatic plant parts [76]. Li et al. [77] estimated the population densities of *P. intermedium* in forest soils using a real-time PCR technique. The primers Pf002 and Pr002b, designed using the ITS region of the *P. intermedium,* were found to be highly species-specific. For the sensitive and quantitative observation of *Pythium* spp., real-time PCR assays were also developed by Li et al. [77], Li et al. [78], and Van der Heyden et al. [79]. 

Very often, multiple *Pythium* spp.are involved simultaneously in plant infections. Multiplex PCR facilitates the simultaneous identification of the species present within a sample. The multiple primers used in a single reaction are designed to hybridize specifically to target DNA and amplify the target regions with different amplicon sizes that can easily be separated by electrophoresis [38,80]. Nine primers—18S-69F, 18S-1118R, AsARRR, AsAPH2B, AsTOR6, AsVANF, AsPyF, AsGRAF, and AsPyR—for identifying five species of *Pythium, P. arrhenomanes, P. graminicola, P. aphanidermatum, P. torulosum,* and *P. vanterpooli* that cause diseases in turfgrass were designed by Asano et al. [80]. In the DNA extracted from plant samples showing symptoms, multiplex PCR was more efficient in detecting the *Pythium* spp. compared to pathogen isolation using the selective medium. Based on the isolation of multiple *Pythium* spp. from one plant sample, it was postulated that *Pythium* spp. would be involved in latent infections, and subsequently, visible symptoms were observed only when the conditions optimal for disease induction occurred. Ishiguro et al. [81] developed a multiplex PCR detection method to identify high-temperature-growing *Pythium* spp., *P. aphanidermatum*, *P. helicoides,* and *P. myriotylum* in soil samples. Species-specific primers—kkMYRR for *P. myriotylum* and kkhel F1mods2/kkhel R2 for *P. helicoides*—were newly designed. Other pre-designed primers—AsPyF for *P. aphanidermatum* and *P. myriotylum,* AsAPH2B for *P. aphanidermatum*—and two universal primers—18S69F and 18S1118R—were also used. Their analysis concluded that the primers were highly species-specific and could efficiently detect specific species from environmental samples.

Loop-mediated isothermal amplification (LAMP), unlike other PCR techniques, does not involve a thermal cycler; instead, nucleic acids are amplified under isothermal conditions [82]. The use of four different primers explicitly designed for six different target gene regions makes it highly specific [83]. Two loop primers were also used in an attempt to reduce the amplification time. The amplification of nucleic acids relies on Bst DNA polymerase, which uses autocycling strand displacement for DNA synthesis [84]. Fukuta et al. [82] developed a LAMP assay to detect *P. myriotylum* detection in hydroponic solution samples. The five primers—F3, B3, FIP, BIP, and B-Loop—designed using the ITS sequence of *P. myriotylum* and related species from the B1 phylogenetic clade of the *Pythium* efficiently amplified the target genes at 60 °C. The DNA extracted from the hydroponic nutrient solution showed the presence or absence of *P. myriotylum* accurately, which was confirmed by the plate culture method. The presence of double-stranded DNA amplified by primers and Bst DNA polymerase was observed by the fluorescence produced due to double-strand binding intercalating dye. As the number of double-stranded DNA increases, fluorescence increases. Additionally, LAMP assays were also developed for *P. irregulare* [85], *P. aphanidermatum* [85], *P. ultimum* [86], and *P. spinosum* [87]. Cao et al. [88] used a modification of LAMP called RealAmp, where LAMP products were quantified in real-time. The quantitative detection of the pathogen was done by measuring the increase in turbidity derived from magnesium pyrophosphate to determine the amplified DNA product. The only limitation associated with LAMP is the false-positive results produced due to the presence of multiple primers, which enhances the sensitivity of the assay manifolds [82]. The LAMP method has a high potential for use in field conditions because of the low requirement for sophisticated instruments; additionally, the environmental samples can be analyzed directly without the need for DNA extraction [83,86]. 

PCR-based diagnosis provides advantages over conventional diagnostic procedures of rapid detection, sensitivity, and the ability to avoid culturing a target pathogen [75]. However, using PCR technology to detect pathogens in environmental samples has its own shortcomings. The extraction of *Pythium* DNA from environmental samples is hindered by the presence of PCR inhibitors, the thick walls of dormant structures, and the strong binding of microbes to soils or plant tissue [37,89,90]. The purity of extracted DNA, the state of infection in plant tissue, and pathogen distribution in environmental samples can also drastically affect the results [38]. PCR amplification cannot provide a delimitation between viable and non-viable propagules because DNA can persist for more extended periods even after cell death, thus hindering the assessment of a control strategy [91]. Using real-time PCR with environmental samples can result in the incorrect quantification of the pathogen in the test sample because of the presence of fluorescent molecules, the difference in the copy number of nuclear rDNA between isolates, and the ambiguity of extracted DNA from living and dead propagules [38,77,92]. Other than the technical limitations, the skill requirements, high-end instruments, and high costs limit the use of PCR-based diagnosis in actual field-based studies.

## 4. Control Measures

### 4.1. Cultural Control

Cultural techniques such as organic amendments, soil solarization, and cover cropping have been observed to control *Pythium*-related diseases in *C. annuum* crops. The solarization of soil has been a prevalent practice to manage various soil-borne diseases in greenhouses and nursery beds. Soil solarization is a pre-sowing management approach that generally reduces colony-forming units (CFUs) or propagules of various soil-inhabiting pathogens as well as other microbes, as observed by Akhtar et al. [93], where CFUs of *Pythium* spp.were reduced to 1.67 × 10^4^ after 8 weeks of solarization as compared to 5.00 × 10^4^ in pre-solarized soils at a depth of 15 cm. 

The use of isothiocyanates (ITCs) producing Brassicaceae cover crops for soil biofumigation has acquired increasing interest as an alternative to restricted chemical fumigants such as methyl bromide for soil-borne disease control [94]. Although isolates of *Pythium* spp. were effectively inhibited by ITCs in in-vitro conditions [95], field experiments using *Brassica* cover for biofumigation were not very efficient in reducing *Pythium* populations in pepper crops [65]. However, high ITC levels were observed in field soils after the pulverization and incorporation of *Brassica* cover crops in soils [65]. In contrast, Handiseni et al. [96] saw an improvement in pepper seedling emergence of 40%–60% when the ITC-producing intact seed meals of *Brassica napus* and *Brassica juncea* were incorporated into *P. ultimum*-infested soils. The difference in observations by Hansen and Keinath [65] and Handiseni et al. [96] can be attributed to the variability in sensitivity to ITC of different species of *Pythium*. 

### 4.2. Chemical Control 

Despite their well-known detrimental effects, such as environmental toxicity and accumulation at different trophic levels [97], the use of chemical pesticides is still relevant because of their effective results. Fungicides used as a soil drenching, fumigation, and seed treatment have been used to control soil-borne disease. Saha et al. [98], in a four-year-long study, studied the effect of the treatment of chili seeds with the fungicidal formulations Thiram 75 WS and Captan 50 WP on incidences of damping-off caused by *P. aphanidermatum*. Treatment of seeds using Thiram 75 WS at the dose of 2.5 kg/g of seeds reduced the pre-emergence damping-off losses from 29.06% in untreated control to 6.52% in treatment and post-emergence damping-off losses from 59.12% in the untreated control to 16.15% in treatment conditions. It was even attempted to integrate metalaxyl with biocontrol agent *Trichoderma harzianum* as a seed treatment, which also produced good results as the percentage of seed rot from 93.75% in untreated seeds was reduced to 52.08% in treated seeds [28]. However, using a single active ingredient has resulted in the development of resistant varieties, as can be seen in metalaxyl-resistant *Pythium* spp. [99,100]. A combination of different fungicides, creating a cocktail of alternative chemistries, has provided the solution to overcome the resistance developed by *Pythium* spp. [99,100]. The two or more active compounds used in this pesticide cocktail mostly act additively, but synergism has also been observed in some cases [101]. It is suggested to use Previcur 840 SL containing propamocarb (47.3%) and fosetyl (27.7%) as a combination of active ingredients, which is recommended for the effective control damping-off disease caused by *Pythium* spp. in chili [102]. The efficacy of mixed fungicides in vitro on isolates of *Pythium* isolated from diseased parts of chili plants was studied by Dubey et al. [21], who observed Vitavax (Carboxin 37.5% + Thiram 37.5%) to be highly effective, inhibiting the growth of mycelia by 93.3% at a concentration of 100 ppm. 

As discussed in the previous sections, high moisture content in the soil is conducive to the development of *Pythium* spp.; thus, events of extreme rainfall can lead to severe disease outbreaks. Saha et al. [103] observed high mortality in the *Capsicum* crop after Hurricane Frances, which was further aggravated due to the biological vacuum created by pre-plant metalaxyl fumigation. The fumigation resulted in a severe reduction in beneficial soil microbiota, which later supported the outbreak by *Pythium* spp. Kokalis-Burelle et al. [104] also observed *Pythium* epidemics in methyl bromide-treated plots of pepper plants in subsequent years, which were attributed to the biological vacuum created by methyl bromide fumigation. Due to the fewer soil microorganisms present in the soil, no competition was present to *Pythium*. Thus, disease forecasting based on variables such as the number of pathogen propagules, the susceptible host, and the conducive environment becomes crucial in deciding the timing and dose of fungicides to be used [105]. 

### 4.3. Biological Control

The antagonistic ability of many fungal, bacterial, and algal isolates has been investigated directly and indirectly against phytopathogens, as shown in Table 2. The activity and efficacy of biocontrol agents on damping-off in seeds and seedlings depends very much on the texture of seeds, the associated microflora in the soil, and the physiological characters of the plant. Therefore, it can be concluded that the efficacy of biocontrol tends to vary from species to species and crop to crop [68]. The antagonism exhibited by microbial biocontrol agents can follow different modes of actions such asc ompetition, induced systemic resistance, antibiosis, and mycoparasitism (Figure 3).

Table 2 shows that *Bacillus* spp. and *Pseudomonas* spp. are the most frequently used and most successful bacterial biocontrol agents in pot trials and field applications. However, both of these have diverse modes of action. Endophytic *Pseudomonas fluorescens* EBS20 isolated from chili plants produced an antibiotic phenazine, which reduced the in vitro growth of *Pythium* spp. [51]. The production of antimicrobial substances by *Pseudomonas* spp. and *Bacillus* spp. was observed by Muthukumar et al. [106] and Amaresan et al. [107], respectively. The disease-suppressing activity of plant growth-promoting rhizobacteria has also been correlated with its efficiency in promoting plant growth [106]. Many species of *Pythium* that are pathogenic to plants have a very high optimum temperature for growth at which some biocontrol agents are not very effective [108]. Mehetre and Kale [109] attempted to assess a thermophile *Bacillus licheniformis* NR1005 for its antagonism against *P. aphanidermatum,* and promising results were obtained in suppressing the disease. 

*Trichoderma* spp. was found to be the most frequently studied fungal biocontrol agent for the control of *Pythium* spp.-related diseases in *Capsicum* plants. *Trichoderma* spp. has been observed to compete with phytopathogens directly by showing a competitive saprophytic ability through the production of cellulase enzymes, thus competing with *Pythium* spp. directly in their ecological niche [112]. The production of antimicrobial substances by *Trichoderma* was also observed to be one of the mechanisms in the control of *Pythium* spp. [112,113]. 

Compatible strains of bacteria and fungi were also observed to inhibit the activity of *Pythium* spp. in vitro and in vivo; using a different mechanism of disease suppression, exhibiting synergistic effects, proved more effective than individual antagonists [52,114]. In the combined treatment of *Pseudomonas fluorescens* EBL 20-PF and *T. viride* TVA on chili plants, Muthukumar et al. [52] observed induced resistance in the plant itself. They noticed increased activities of resistance-inducing enzymes and the accumulation of phenolic compounds, resulting in a significant decrease in disease incidences. 

Non-native antagonist isolates generally face resistance by pre-established and acclimatized microflora and often fail to establish themselves [107]. Antagonists isolated from the plant rhizosphere and from the plant itself have shown convincing results in reducing incidences of damping-off [51,53,113]. However, as previously discussed, propagules of *Pythium* spp. can germinate and infect very rapidly, sometimes even with a shortage of nutrients and water, thus escaping natural antagonism [58]. An antagonist can only be efficient against *Pythium* spp. if provided with an advantage in colonizing the roots and rhizosphere of the plant. Early inoculation of the biocontrol agent before seeding or inoculation close to the seed or seedling can increase the efficacy of an antagonist [115]. The rapid colonization of an antagonist can also provide an extra advantage for its success [109,110]. 

## 5. Virulence Mechanism of *Pythium* spp. and Challenges in Resistant Breeding against *Pythium* spp.

Disease development by *Pythium* spp. involves an extensive repertoire of carbohydrate-active enzymes (CAZymes), including glycoside hydrolases, polysaccharide lyases, carbohydrate esterases, proteases, etc., which help in plant cell wall penetration and further colonization [116,117]. Lévesque et al. [117], in their study of the *P. ultimum* genome, observed high sequence similarity and synteny with another phytopathogenic oomycete, *Phytophthora*, sharing genes and encoding enzymes involved in the metabolism of carbohydrates. However, the absence or underrepresentation of many notable genes in *Pythium* encoding for cutinases, xylanases, pectinases, etc., that are present in the *Phytophthora* genome suggests some difference in the methods of virulence between these oomycetes [118]. Even the expression of cellulases and pectinases in *Pythium* spp. are limited; they are sufficient for hyphal penetration but not for complete saccharification. Furthermore, the absence of these complex carbohydrate-degrading enzymes and the presence of α-glucosidase, α-amylases, α-glucoamylases, and invertases, degrading the plant starch and sucrose, establishes the affirmation of phytopathogenic *Pythium* towards young plant tissues with no secondary growth [116]. The presence of genes encoding proteases families such as subtilisin-related proteases, metalloproteases, and E3 ligases, inducing necrosis and cell-wall degradation in the core *Pythium* spp. genes and the absence or lack of RXLR effectors in *Pythium* support its non-host specificity and necrotrophic lifestyle [117,118,119]. Unlike *Phytophthora*, *Pythium* spp. have been observed to lack RXLR effectors with avirulence activities [118,119]. The absence of these avirulence factors has been correlated with the lack of gene-for-gene resistance against *Pythium* spp. Thus, the absence of a virulence factor producing RXLR effectors makes the identification of race-specific resistance genes against *Pythium* very difficult to explore [120]. However, Ai et al. [121] have recently identified RXLR effectors in nine *Pythium* spp. that share a common ancestor with *Phytophthora*. RXLR effectors in *Pythium* spp. exhibited necrosis-inducing activities resulting in plant cell death. The difficulties in identifying race-specific resistance genes against *Pythium* spp. make quantitative gene expression a more comfortable approach for developing resistant varieties. Several major and minor quantitative trait loci (QTL) associated with root rot resistance have been identified in *Pythium* spp.-affected crops such as snap bean [122] and soybean [120,123,124,125]. The partial (horizontal) resistance achieved through the involvement of multiple QTLs in the soybean plants leading to transgressive segregation has been observed to confer a common response pattern against multiple *Pythium* spp. [120,125]. As *Pythium* spp.-affected crops are often inhabited by several *Pythium* spp., the development of a variety with resistance against multiple species would be more efficient. 

## 6. Conclusions

As one of the significant horticulture crops in India and worldwide, improvements in *Capsicum* production yield are a primary target that needs to be achieved. Biotic stress due to different soil-borne pathogens poses a significant threat to that target. Different *Pythium* spp. have been reported to cause infections in *Capsicum* cultivars with a disparate reproductive behavior and growth environment. These factors make the detection of the pathogen and correct identification of the species of paramount importance. The technology for the detection and diagnosis of pathogens has evolved from polyclonal antibodies to the use of multiple species-specific primers for a single species. The use of LAMP technology has provided a possible escape from the use of sophisticated instruments and pure DNA isolation. However, the search for a point-of-care technology that does not require a specific skill set and can provide results in a shorter time frame is still far from over. 

Although biological control has provided a viable alternative to chemical management, in a natural setup, the comparative disease control of *Pythium* spp. has not been achieved yet. Additionally, the successful expression of antagonism by the microbes partially depends upon the cultural and environmental conditions. An integrated system of cultural control and microbial control can provide the desired targets in *Pythium* spp. disease control. The use of disease-resistant varieties is another sustainable alternative for disease control. However, in this case, no research work could be found involving *Pythium*-resistance development in *C. annuum* cultivars. The candidate QTL genes for the resistance against multiple *Pythium* spp. could prove to be groundbreaking in resistance breeding in *C. annuum* L.

## Figures and Tables

**Figure 1 microorganisms-09-00823-f001:**
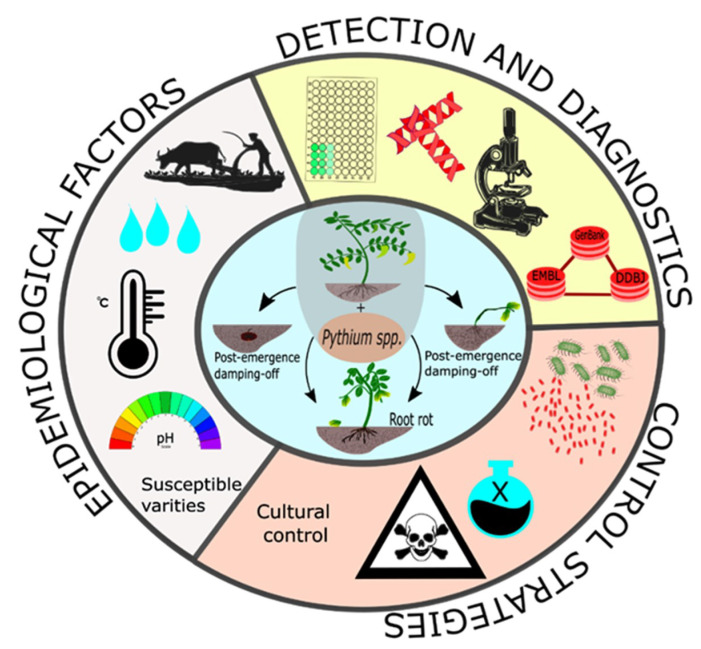
Pictorial representation of the components reviewed in this study.

**Figure 2 microorganisms-09-00823-f002:**
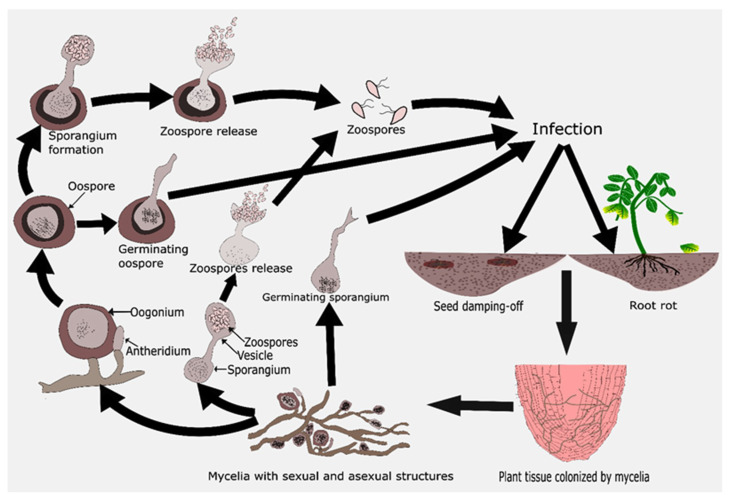
Disease cycle of *Pythium* species in *Capsicum* plants.

**Figure 3 microorganisms-09-00823-f003:**
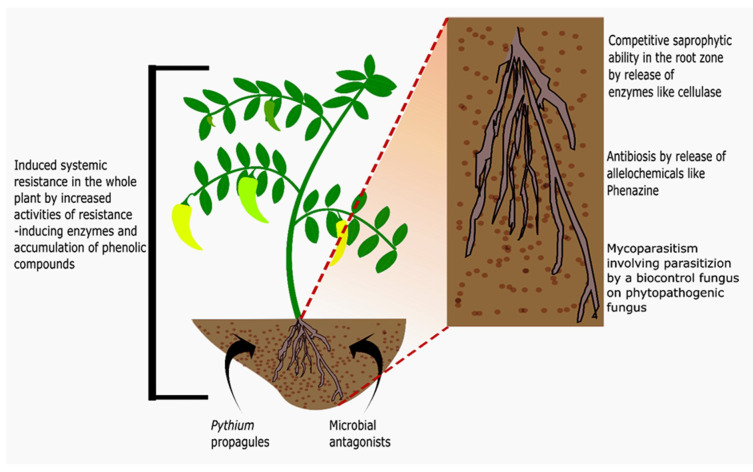
Localized antagonism shown by microbes in the soil against *Pythium* spp. through competition, antibiosis, and mycoparasitism, and localized and delocalized antagonism by microbes as a result of induced systemic resistance.

**Table 1 microorganisms-09-00823-t001:** Reported *Pythium* species associated with diseases in *Capsicum annuum* L.

Country	Species	Identification Criteria	Sequence Used in Molecular Identification	Crops Affected	Reference
Pakistan	*P. myriotylum*	Morphological and molecular	ITS sequence	13.8%–45.4%	[20]
India	*P. graminicola*	Morphological and molecular	ITS sequence	Not available	[21]
Pakistan	*P. spinosum*	Morphological and molecular	ITS sequence	Not available	[23]
India	*P. aphanidermatum*, *P. graminicola*, *P. ultimum*, *P. diliense, P. heterothallicum*	Not available	Not available	Not available	[26]
Pakistan	*P. debaryanum*	Morphological and molecular	ITS sequence	45%	[32]
India	*P. aphanidermatum*	Morphological	Not available	Not available	[48]
Pakistan	*P. aphanidermatum*	Morphological and molecular	ITS and partial LSU sequence	Not available	[49]
Pakistan	*P. aphanidermatum* *P. spinosum* *P. intermedium*	Morphological and molecular	ITS Sequence	Not available	[50]
India	*P. aphanidermatum*	Not available	Not available	Not available	[51]
India	*P. aphanidermatum*	Not available	Not available	Not available	[52]
Pakistan	*Pythium* spp.	Not available	Not available	Not available	[53]

ITS: Internal Transcribed spacer; LSU: Large subunit ribosomal DNA.

**Table 2 microorganisms-09-00823-t002:** Exploration of microbial species for *Pythium* control in *Capsicum annuum* (from 2010 to present).

Pathogen Species	Microbial Control	Strain Name/Commercial Product	In Vitro Control	In Vivo Control	Mode of Action	Reference
Bacteria						
*P. aphanidermatum*	*Pseudomonas fluorescens*	EBS20	76.66% reduction in the growth of mycelia.	Not available	Production of phytopathogen inhibitor phenazine.	[51]
*P. aphanidermatum*	*Pseudomonas. fluorescens*	Biomonas *	Not available.	10.46% and 20.28% losses due to pre and post-emergence damping-off, respectively, as opposed to 29.06% and 59.12% in control in nursery fields using seed treatment.	Not available.	[98]
*P. aphanidermatum*	*Pseudomonas fluorescens*	EBC 5	68.88% reduction in the growth of mycelia.	9.10% and 12.33% incidences of pre and post-emergence damping-off when EBC 5 and EBC 7 were combined in pot culture using seed coating, as opposed to 30.66% and 34% in control.	Production of antifungal metabolites reduced mycelial growth in-vitro.	[106]
	*Pseudomonas fluorescens*	EBC 7	65.93% reduction in the growth of mycelia.			
*Pythium* spp.	*Bacillus megaterium*	BECS7	45.9% reduction in the growth of mycelia.	2% incidences of damping-off as opposed to 14.67% in control in field conditions.	Release of hydrolytic enzymes such as lipase, cellulase, amylase, and protease.	[107]
*P. aphanidermatum*	*Bacillus licheniformis*	NR1005	69.96% reduction in the growth of mycelia.	81.18% reduction in damping-off incidence over control in pot culture using seed treatment.	Not available.	[109]
*P. ultimum*	*Stenotrophomonas rhizophila*	KM01	80% reduction in the growth of mycelia.	75%–100% reduction in disease index over control in pot culture using root inoculation.	Not available.	[110]
	*Stenotrophomonas rhizophila*	KM02	76% reduction in the growth of mycelia.	75%–100% reduction in disease index over control in pot culture using root inoculation.		
	*Bacillus subtilis*	RBM02	67%–77% reduction in the growth of mycelia.	100% reduction in disease index over control in pot culture using root inoculation.		
*P. debaryanum*	*Bacillus subtilis*	RB-31	91% inhibition of mycelial growth.	Not available.	Not available.	[111]
Fungi						
*Pythium* spp.	*Trichoderma harzianum*	TK8	62.8% reduction in the growth of mycelia.	Not available.	Not available.	[29]
*P. aphanidermatum*	*Trichoderma harzianum*	Not available	75.34% reduction in the growth of mycelia.	83.16% reduction in damping-off incidence over control in pot culture using seed treatment.	Not available.	[109]
*P. ultimum*	*Cryptococcus laurentii*	2R1CB	75% reduction in the growth of mycelia.	75%–100% reduction in disease index over control in pot culture using root inoculation.	Production of β-1,3-glucanase reduced the mycelial growth in vitro.	[110]
*P. aphanidermatum*	*Trichoderma viride*	Not available	76.1% reduction in the growth of mycelia.	Not available	Production of antibiotics.	[112]
*P. aphanidermatum*	*Trichoderma viride*	TVC_3_	88% reduction in the growth of mycelia.	Not available	Volatile and non-volatile antibiotics production and mycoparasitism.	[113]
Fungi + Bacteria						
*P. aphanidermatm*	*Trichoderma viride* + *Trichoderma harzianum + Pseudomonas fluorescens + Bacillus* *subtilis*	Not available.	Not available.	13.33% and 15.36% incidences of pre and post-emergence damping-off, respectively, as opposed to 53.33% and 24.80% in control in pot culture using seed treatment.	Not available.	[30]
*P. aphanidermatum*	*Trichoderma viride* + *Pseudomonas fluorescens*	TVA EBL 20-PF	Not available.	Reduction of 84% and 71.5% in pre and post-emergence damping-off incidences, respectively, using seed treatment and soil application in pot culture.	Induced systemic resistance due to increased activities of PAL, PO, PPO, and accumulation of phenolics.	[52]
*P. aphanidermatum*	*Trichoderma viride* + *Pseudomonas fluorescens*	Not available.	82% reduction in mycelial growth over control.	Reduction of 72.2% and 59.2% in pre and post-emergence damping-off incidences, respectively, in pot culture, using seed treatment.	Production of antifungal antibiotic.	[114]
Algae						
*P. aphanidermatum*	*Calothrix elenkenii*	Not available.	Minimum inhibitory concentration of the ethyl acetate extract of culture filtrate was 16.6 ppm.	Seed treatment with ethyl acetate extract of culture filtrate reduced mortality to 10%–20% as opposed to 60%–70% in untreated controls in pot culture.	Not available.	[63]

* Commercial formulation product of *Pseudomonas fluorescens.*

## Data Availability

All the data is present in the manuscript.

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
