# Peer review of "Pythium* Damping-Off and Root Rot of *Capsicum annuum* L.: Impacts, Diagnosis, and Management"

_microorganisms, 2021, doi:10.3390/microorganisms9040823_

Round 1
Reviewer 1 Report
In my opinion, the work is very well written. It is a good study of two of the most important diseases of peppers caused by Pythium. The authors presented all aspects related to the discussed issue, ranging from the symptoms of diseases and the losses caused, through the methods of pathogen detection, to the protection strategies, both taking into account chemical and biological methods.
after reading the work, I have only one comment. namely, lines 129-130 - the authors list a number of diseases caused by Pythium but do not cite any literature references. In my opinion, it is good to support these types of examples with publications.
Reviewer 1 Report
In my opinion, the work is very well written. It is a good study of two of the most important diseases of peppers caused by Pythium. The authors presented all aspects related to the discussed issue, ranging from the symptoms of diseases and the losses caused, through the methods of pathogen detection, to the protection strategies, both taking into account chemical and biological methods.
after reading the work, I have only one comment. namely, lines 129-130 - the authors list a number of diseases caused by Pythium but do not cite any literature references. In my opinion, it is good to support these types of examples with publications.
Response: Authors are thankful to the reviewer for the kind appreciation. As suggested, the literature references have been added in the revised manuscript [Line No.194].
Reviewer 2 Report
I am pleased to read this manuscript. Here are my comments and suggestions:
The abstract does not refer to the form recommended by the MDPI journal. It should include an introduction, methods, results and conclusions. Meanwhile, in its present form, it is mainly an encyclopedic description of the species, unfortunately wrong. Because Capsicum annum L. is a species known all over the world for both sweet and spicy fruits. The fruit of the sweet variety is used as a vegetable, the fruit of the spicy variety is used as a cooking spice.
Moreover, decay and root rot caused by the pathogen Pythium spp., Whose habitat is soil, is a problem for many horticultural plants, both vegetables and flowers, in their initial growth period. This topic is very well documented in the various studies available for over 60 years.
One would rather expect that this study will concern a specific method of cultivation, because in most countries of the world paprika is an annual plant grown from seedlings, while in tropical countries it is a perennial and the problem of pathogen remaining in the soil is a big problem.
The introduction is not well organized, some of the content is basic information about the species and type and various forms of fruit shape. The description of the chemical composition is not very informative, as is Chapter 2. Economic Impacts of damping-off and root rot disease on C. annuum L.
The literature review of the pathogen does not cover most of the studies related to the plant species. Summary based on the thesis that ... "Suppression of rot and root rot caused by Pythium spp. Are part of a threat that largely goes undetected due to a tendency to infect juvenile tissue, often mistaken for poor seed quality or other abiotic factors. . "... is wrong. Experienced growers and growers can distinguish the initial symptoms of the disease very well and counteract them effectively.
Moreover, the cultivation of sweet pepper is not the most important crop in the world. There are many other types of vegetables that are grown on a larger scale. This comparison can only apply to countries such as India, Hungary and Bulgaria. In the rest of the world they are marginal crops.
More than 1/3 of the journals concern research and reports from 10 years ago. Most of them are textbook studies.
All this means that in its ts current form, the article does not meet the requirements of a critical review, which should be balanced.
Author Response
Reviewer 2 Comments
Comment 1: The abstract does not refer to the form recommended by the MDPI journal. It should include an introduction, methods, results, and conclusions. Meanwhile, in its present form, it is mainly an encyclopedic description of the species, unfortunately wrong. Because Capsicum annum L. is a species known all over the world for both sweet and spicy fruits. The fruit of the sweet variety is used as a vegetable, the fruit of the spicy variety is used as a cooking spice.
Response: Authors are thankful to the reviewer for the kind suggestion. The current manuscript does not fall under the research article where methods and results constitute a significant part of the abstract. However, a conclusion was missing in our abstract, and the same has now been added (Line No. 23-28).
Yes, we do agree with the comment of the reviewer that Capsicum annuum L. includes both sweet and pungent (spicy) fruits. In order to avoid discrepancy, we now have deleted the word “sweet” variety from the manuscript. Thanks again for providing meaningful comments.
Comment 2: Moreover, decay and root rot caused by the pathogen Pythium spp., Whose habitat is soil, is a problem for many horticultural plants, both vegetables, and flowers, in their initial growth period. This topic is very well documented in the various studies available for over 60 years. One would rather expect that this study will concern a specific method of cultivation because in most countries of the world paprika is an annual plant grown from seedlings, while in tropical countries it is a perennial and the problem of pathogen remaining in the soil is a big problem.
Response: In all agreement with the learned reviewer, the current manuscript focuses on the advancement in detection and diagnosis techniques of Pythium spp., management approaches attempted in Capsicum plants and the scope of development of resistant varieties for the annual cropping system of C. annuum L. In order to clear the confusion, we have incorporated a statement (line No.37-39) mentioning the same. Thanks to the reviewer for highlighting the issue.
Comment 3: The introduction is not well organized, some of the content is basic information about the species and type and various forms of fruit shape. The description of the chemical composition is not very informative, as is Chapter 2. Economic Impacts of damping-off and root rot disease on C. annuum L.
Response: The introduction has been revised as per the kind suggestion of the reviewer particularly the chemical composition of the C. annuum (Line No. 50-52, 55-61).
Lines 70-71 from the original manuscript were not in line with the flow of the introduction and these lines have been removed. Likewise, chapter 2 has now been edited and merged with chapter 1 itself. Thanks for the suggestion.
Comment 4: The literature review of the pathogen does not cover most of the studies related to the plant species. Summary based on the thesis that ... "Suppression of rot and root rot caused by Pythium spp. Are part of a threat that largely goes undetected due to a tendency to infect juvenile tissue, often mistaken for poor seed quality or other abiotic factors. . "... is wrong. Experienced growers and growers can distinguish the initial symptoms of the disease very well and counteract them effectively. Moreover, the cultivation of sweet pepper is not the most important crop in the world. There are many other types of vegetables that are grown on a larger scale. This comparison can only apply to countries such as India, Hungary, and Bulgaria. In the rest of the world, they are marginal crops.
Response: As mentioned in comment 1, the manuscript now focuses on pungent fruit, and all the references and statements pertaining to sweet pepper have now been removed. The summary “Suppression of…...abiotic factors” was concluded from the studies available in the literature. However, keeping in mind our respected reviewer’s expertise in this field, the relevant statements in the originally submitted manuscript (Line No. 77-79, 427-430) have been removed in the revised manuscript.
Comment 5: More than 1/3 of the journals concern research and reports from 10 years ago. Most of them are textbook studies.
Response: The current study, considering the host-pathogen relationship between C. annuum L. and Pythium, is the first of its kind. In an exhaustive search, few studies could be found from the last 2-3 years relevant to the topic, proving this host-pathogen system to be an underexplored domain. So, to establish a firm opinion, an ample amount of the research had to be kept, and up-to-date studies relevant to this topic have been mentioned in the text.
Still, during re-editing of the manuscript (while addressing comments No. 1-4), many older references have been removed. Currently, the review has 113 references compared to 128 references in the original manuscript. Thank you so much for making the review crisp and contemporary.
Comment 6: All this means that in its current form, the article does not meet the requirements of a critical review, which should be balanced.
Response: Thanks for your critical assessment of our manuscript. We now believe that after applying all your suggestions, the review has now become balanced and updated.
Reviewer 3 Report
The manuscript titled “Pythium damping-off and root rot of Capsicum annuum L.: Impacts, Diagnosis, and Management” is devoted to damping-off and root rot caused by a soil-borne pathogen Pythium spp. in Capsicum plants production.
It is a review of 128 published references about Economic Impacts of damping-off and root rot disease on C. annuum L.; short review of Pythium species: as causal agents of the Damping-off and Root rot; some words about detection and diagnosis of the species Pythium; Control Measures (Cultural Control, Chemical Control, Biological Control); and Virulence mechanism of Pythium and source of resistant breeding in Capsicum against Pythium.
Unfortunately, Reviewer has opinion that this manuscript in general does not fit to the aim of MDPI Microorganisms journal: “… to encourage scientists to publish their experimental and theoretical results in as much detail as possible.” The Authors chose another way – to give very broad picture without details.
The review must be rather concentrated on one of chosen topics, and deep analysis of published with theories and conclusions.
Additionally, an extensive editing of English language and style is advised before next submission of the manuscript.
Reviewer 3 Report
The manuscript titled “Pythium damping-off and root rot of Capsicum annuum L.: Impacts, Diagnosis, and Management” is devoted to damping-off and root rot caused by a soil-borne pathogen Pythium spp. in Capsicum plants production.
It is a review of 128 published references about Economic Impacts of damping-off and root rot disease on C. annuum L.; short review of Pythium species: as causal agents of the Damping-off and Root rot; some words about detection and diagnosis of the species Pythium; Control Measures (Cultural Control, Chemical Control, Biological Control); and Virulence mechanism of Pythium and source of resistant breeding in Capsicum against Pythium.
Unfortunately, Reviewer has opinion that this manuscript in general does not fit to the aim of MDPI Microorganisms journal: “… to encourage scientists to publish their experimental and theoretical results in as much detail as possible.” The Authors chose another way – to give very broad picture without details.
The review must be rather concentrated on one of chosen topics, and deep analysis of published with theories and conclusions.
Additionally, an extensive editing of English language and style is advised before next submission of the manuscript.
Response: Thanks for highlighting the major flaws of the manuscript. The complete manuscript from Abstract to Conclusion has now been re-edited. The review in its revised form has become more contemporary and precise, as evident from the number of references that have been narrowed down to 113 from the original 128.
Regarding the language, the manuscript has been checked from “Grammarly” (Premier version) before submission.
Round 2
Reviewer 2 Report
The authors re-edited the manuscript. They revised the manuscript thoroughly in line with the comments of the reviewer. They took into account and addressed all suggestions. Doubts have been fully explained. In my opinion, the manuscript in its current form may be published.
Author Response
Authors Response to Reviewer 2 Round 2 Report
- The authors re-edited the manuscript. They revised the manuscript thoroughly in line with the comments of the reviewer. They took into account and addressed all suggestions. Doubts have been fully explained. In my opinion, the manuscript in its current form may be published.
Authors’ response: The authors are thankful to the reviewer for the excellent evaluation and reviewing of the manuscript. His suggestions have helped in improving the MSS to an excellent shape.
The authors thanks to the reviewer for ENDORSING the publication of the MSS in it current form.
Reviewer 3 Report
In my previous evaluation, I advised to concentrate on something more specific (taxonomy, virulence, disease control, etc.) and make a deeper analysis of published data. So far, I do not see any significant improvements in the manuscript, besides some cosmetic changes. I cannot recommend accepting the manuscript without changes, but, I do advice several changes. First, to reduce description of crop given at lines 42-80. It is not related to the manuscript aim. The manuscript titled "Pythium damping-off and root rot of Capsicum annuum L.: Impacts, Diagnosis, and Management" Thus, it is important to describe more carefully diagnostic methods (including primers for PCR and LAMP) of the pathogen detection, to show major taxonomic groups, their geographic distribution, and problems of their differentiation. Part devoted to management of the pathogen by chemical methods is too short and does not address such important problems as pathogen resistance to pesticides, application of pesticides mixtures, forecasting pathogen development based on climate, soil and crop properties. Biocontrol part has a lot of detailes (Table 3. Exploration of microbial species for Pythium control in Capsicum annuum (From 2010 to present)), but does not show availability of applied biocontrol agents (name of strain, microbial collection or commercial product). Instead, it describes particular effects achieved in the some experiments (in many cases with unnamed isolates of species) - this effect is often difficult to reproduce in other experiment. The part of review devoted to plant resistance has very strange conclusion: "Since now, no studies could be observed regarding resistance development in Capsicum against Pythium spp. However, presence of resistant quantitative genes against another oomycete in Capsicum does provide a target source of genetic resistance which can be exploited." Thus, this part "Virulence mechanism of Pythium spp. and source of resistant breeding in Capsicum against Pythium spp." is very speculative, and can be reduced to the above cited sentence. I would advice to re-write the review with suggested changes.Author Response
Reviewer 3 Round 2 Report
- In my previous evaluation, I advised to concentrate on something more specific (taxonomy, virulence, disease control, etc.) and make a deeper analysis of published data. So far, I do not see any significant improvements in the manuscript, besides some cosmetic changes.
Authors’ response: With all due respect to the reviewers’ critical assessment of the manuscript, the authors have attempted to re-revise the manuscript in a more specific manner.
- First, to reduce description of crop given at lines 42-80. It is not related to the manuscript aim. The manuscript titled "Pythium damping-off and root rot of Capsicum annuum L.: Impacts, Diagnosis, and Management" Thus, it is important to describe more carefully diagnostic methods (including primers for PCR and LAMP) of the pathogen detection, to show major taxonomic groups, their geographic distribution, and problems of their differentiation.
Authors’ response: Thanks for the kind suggestion. From the introduction, lines 42-43, 55-58, and 67-68 have been removed from the previously revised manuscript. However, the section pertaining to the chemistry of the host plant has to be kept, as it was further elaborated upon after the suggestion of another reviewer.
In the section “Detection and diagnosis of Pythium species,” the authors have attempted to address all diagnostic techniques in a more mannered way, incorporating the problems associated with species identification and differentiation, delineating feature between the Pythium clades, target nucleotide sequence for primer designing and limitation of PCR-based techniques. Line No. 179-185, 193-206, 208-209-228, 213-215, 219-221, 228-244, 249-251, 252-254, 256-259, 262-264, and 269-284 have been added in the revised manuscript, keeping in mind the suggestions given by our respected reviewer.
- The part devoted to the management of the pathogen by chemical methods is too short and does not address such important problems as pathogen resistance to pesticides, application of pesticides mixtures, forecasting pathogen development based on climate, soil, and crop properties.
Authors’ response: Following the enriching suggestion given by our respected reviewer, the issues related to pesticide resistance, predominantly metalaxyl-resistance in the case of Pythium spp. and use of mixed fungicides as a solution have been raised. The moisture content of the soil provides a conducive environment for Pythium spread and growth. Fungicide treatment was observed to supplement the Pythium spread after the heavy rainfall due to the biological vacuum created by the fungicide. Addressing all these suggestions, section 4.2 (chemical control) has been revised, and Line no. 309-311, 320-328, and 333-345 have been added in the revised manuscript.
- Biocontrol part has a lot of detailes (Table 3. Exploration of microbial species for Pythium control in Capsicum annuum (From 2010 to present)), but does not show availability of applied biocontrol agents (name of strain, microbial collection or commercial product). Instead, it describes particular effects achieved in the some experiments (in many cases with unnamed isolates of species) - this effect is often difficult to reproduce in other experiment.
Authors’ response: As per the reviewers’ suggestion, strain name and commercial product name have been added in a separate column in Table 3. However, for five of the biocontrol agents, no source details were mentioned in the source papers themselves (and same has been mentioned in the manuscript too).
- The part of review devoted to plant resistance has very strange conclusion: "Since now, no studies could be observed regarding resistance development in Capsicum against Pythium spp. However, presence of resistant quantitative genes against another oomycete in Capsicum does provide a target source of genetic resistance which can be exploited." Thus, this part "Virulence mechanism of Pythium spp. and source of resistant breeding in Capsicum against Pythium spp." is very speculative, and can be reduced to the above cited sentence.
Authors’ response: The authors highly appreciate the reviewers’ keen observation. The putative conclusion drawn from the literature, which as per the reviewers’ knowledge and authors’ re-evaluation found to be very speculative, has now been removed (Line No. 373-376, 397-399). Also, the section title has been re-titled to “Virulence mechanism of Pythium spp. and challenges in resistant breeding against Pythium spp.” and line No. 420-422 and 424-426 have been added to streghtened the review.
- I would advice to re-write the review with suggested changes.
Authors’ response: With the valuable suggestions given by our respected reviewer, the manuscript has been revised. Authors’ believe that the manuscript has ameliorated remarkably after the current improvements done with the help of the reviewers’ critical evaluation. Thanks.